# Identification of NOX4 as a New Biomarker in Hepatocellular Carcinoma and Its Effect on Sorafenib Therapy

**DOI:** 10.3390/biomedicines11082196

**Published:** 2023-08-04

**Authors:** Hui-Zhou Li, Qing-Qing Liu, De-Hua Chang, Shu-Xian Li, Long-Tao Yang, Peng Zhou, Jiang-Bei Deng, Chang-Hao Huang, Yu-Dong Xiao

**Affiliations:** 1Department of Radiology, The Second Xiangya Hospital, Central South University, Changsha 410011, China228211092@csu.edu.cn (S.-X.L.);; 2Department of Oncology, Xiangya Hospital, Central South University, Changsha 410008, China; 3Department of Diagnostic and Interventional Radiology, University Hospital Heidelberg, 69120 Heidelberg, Germany; 4Department of Pathology, The Second Xiangya Hospital, Central South University, Changsha 410011, China; 5Department of Intervention, Changsha Central Hospital, University of South Chian, Changsha 410011, China; 6The Hunan Provincial Key Laboratory of Precision Diagnosis and Treatment for Gastrointestinal Tumor, Xiangya Hospital, Central South University, Changsha 410008, China

**Keywords:** hepatocellular carcinoma, NOX4, epithelial–mesenchymal transition, sorafenib resistance

## Abstract

To improve the survival of patients with hepatocellular carcinoma (HCC), new biomarkers and therapeutic targets are urgently needed. In this study, the GEO and TCGA dataset were used to explore the differential co-expressed genes and their prognostic correlation between HCC and normal samples. The mRNA levels of these genes were validated by qRT-PCR in 20 paired fresh HCC samples. The results demonstrated that the eight-gene model was effective in predicting the prognosis of HCC patients in the validation cohorts. Based on qRT-PCR results, NOX4 was selected to further explore biological functions within the model and 150 cases of paraffin-embedded HCC tissues were scored for NOX4 immunohistochemical staining. We found that the NOX4 expression was significantly upregulated in HCC and was associated with poor survival. In terms of function, the knockdown of NOX4 markedly inhibited the progression of HCC in vivo and in vitro. Mechanistic studies suggested that NOX4 promotes HCC progression through the activation of the epithelial–mesenchymal transition. In addition, the sensitivity of HCC cells to sorafenib treatment was obviously decreased after NOX4 overexpression. Taken together, this study reveals NOX4 as a potential therapeutic target for HCC and a biomarker for predicting the sorafenib treatment response.

## 1. Introduction

The incidence of liver cancer is anticipated to exceed one million cases by 2025, thus continuing to be a major global health concern [1]. Hepatocellular carcinoma (HCC), the most common malignancy in liver cancer, exhibits a high heterogeneity and risk of metastasis and its treatment depends on tumor progression [2]. The three main treatments for early-stage HCC include surgical resection, liver transplantation, and local ablation [3]. However, tumor recurrence and metastasis occur in up to 70% of patients who received local ablation or resection five years after treatment, significantly reducing the survival and quality of life for HCC patients [4]. In addition, HCC is often diagnosed at an advanced stage when potentially curative therapies are least effective. With high rates of recurrence and drug resistance, medical therapies such as chemotherapy, chemoembolization, and sorafenib therapy continue to be unsatisfactory for these patients. Timely diagnosis and treatment will be critical for the improvement of patients’ survival.

Previous studies have found that abnormal gene expression is strongly associated with the biological aggressiveness of HCC and patient survival. This has generated strong interest in elucidating corresponding prognostic and predictive biomarkers in order to improve outcomes through better selection of patients for anti-cancer therapy. Prognostic biomarkers can provide information about the overall prognosis of a cancer patient, independent of treatment, while predictive biomarkers can provide information about the effects of therapeutic interventions [5]. Prognostic biomarkers may be particularly useful as predictive markers, therapeutic targets, and hypothesis tests of relevance in selecting patients for adjuvant therapy. Therefore, it is essential to identify effective molecular targets to improve patient survival and quality of life.

Given the impressive developments of high-throughput technologies, The Cancer Genome Atlas (TCGA) (https://portal.gdc.cancer.gov/, accessed on 1 August 2022) databases and the Gene Expression Omnibus (GEO) (https://www.ncbi.nlm.nih.gov/gds, accessed on 1 August 2022) have been established to provide a wealth of high-quality data on HCC, which prompted us to consider whether HCC (rather than matched non-tumor) tissue data could be used to create a gene-based risk score and find effective targets to improve the prognosis of HCC. The HCC gene expression profile dataset from the GEO and TCGA databases was examined first in this research. An eight-gene risk model was developed and its association with survival in HCC patients was evaluated. In addition, we examined key genes in collected clinical samples and identified a brand-new biomarker called NOX4 based on qRT-PCR results. The membrane-bound NADPH oxidases of the NOX family are a major source of reactive oxygen species (ROS) in cancer [6]. Multiple NOX members have been found to be dysregulated in multiple cancer models, with NOX4 being the most frequently expressed member [7]. Numerous studies have shown that NOX4 plays a crucial role in tumorigenesis and tumor progression by supporting cancer cell transformation, proliferation, migration, invasion, and epithelial–mesenchymal transition (EMT) [8]. However, whether NOX4 was involved in the progression of HCC remained a mystery. Subsequently, this study further investigated the relationship between NOX4 expression in HCC and patients’ survival, as well as the effects of NOX4 over/down expression in vitro migration, proliferation, and in vivo tumor growth. In addition, effective predictive biomarkers to identify HCC patients who would benefit from systemic therapy remain limited [9]. This study further analyzed the relationship between NOX4 expression and systemic treatment response in HCC. The flow chart is shown in Figure 1.

## 2. Materials and Methods

### 2.1. Data Acquisition from the Cancer Genome Atlas (TCGA) Databases and Gene Expression Omnibus (GEO)

Overall, 375 HCC samples and 32 normal samples data (including the clinical data, mRNA expression profiles, and somatic mutation information) were retrieved from the TCGA, GSE14520and GSE20140 data sets were acquired from the GEO. Then, the counts matrix from the Xena database was downloaded (https://xena.ucsc.edu/, accessed on 1 August 2022). We used the Conbat algorithm to correct batch effects.

### 2.2. Differentially Expressed Genes and Functional Annotation and Gene Enrichment Analysis

We used the limma package to analyze differentially expressed genes (DEGs) between tumors and normal tissues with *p*-values < 0.05 and logFC > 1 as follows. To identify the biological mechanism underlying alterations in HCC, we enriched these DEGs sequentially with Gene Ontology (GO) and Kyoto Encyclopedia of Genes and Genomes (KEGG) using the clusterProfiler tool. The “c2.cp.kegg.v6.2.symbols” dataset from the MSigDB database served as our reference gene set and we utilized GSEA to identify pathways that were either activated or inhibited. Benjamini–Hochberg was the statistical approach employed and the *p*-values were graded.

### 2.3. Lasso Regression Model Construction and the Nomogram Prediction Model

To search for genes related to tumor prognosis, we applied the Least Absolute Shrinkage and Selection Operator (LASSO) Cox regression technique. The appropriate λ was validated using 10× cross-validation. The selected genes were scored using the formular risk factor ∑ = ∗ = risk score Coef Expri i. Cox regression modeling and norm plots were then performed using the selected genes and clinical prognostic data from TCGA.

### 2.4. Patients and Specimens

In total, 150 HCC patients were initially treated with liver resection from the Second Xiangya Hospital of Central South University between August and December 2013, and paraffin-embedded samples were gathered. The process for the calculation of sample size was illustrated in the section on statistical analysis. The clinicopathological characteristics of the patients are summarized in Table 1. Follow-up was performed via telephone interviews (February 2022) or at the last visit to the hospital if a telephone interview was unavailable. For the analysis of NOX4 mRNA and protein levels, we additionally collected an additional 20 pairs of fresh frozen samples from HCC patients who underwent liver resection at our hospital between August and November 2021. HCC tissues were immediately preserved at −80 °C for RNA and protein extraction. This study was approved by the Ethics Committee of the Second Xiangya Hospital. Written informed permission was not required for the 150 patients between August and December 2013 because the study was retrospective in nature. Between August and November 2021, 20 additional participants provided written informed consent.

### 2.5. Immunohistochemical Staining and Evaluation

There were 150 paraffin-embedded HCC tissues used in total. We performed immunohistochemical (IHC) staining to detect NOX4 protein expression in HCC samples. Briefly, 5 μm sections of paraffin-embedded samples were sliced and a gradient of xylene and alcohol was used to dehydrate them. Antigens were extracted for 15 min in a solution of 100 mmol/L sodium citrate (pH 6.0) and then chilled for 30 min. After inhibition of endogenous peroxidase activity with 3% hydrogen peroxide, nonspecific binding was blocked with 10% goat serum. Sections were incubated overnight at 4 °C with anti-NOX4 and anti-Arginase-1 and anti-Glypican-3 antibodies overnight at 4 °C (mouse polyclonal antibody,1:100 dilution). The samples were then incubated with biotinylated secondary antibody for 30 min at 37 °C and stained with DAB (3,3-diaminobenzidine) and Mayer’s hematoxylin. The expression of NOX4 was assessed semi-quantitatively according to the intensity of staining (score 0, no staining; score 1, weak staining; score 2, moderate staining; score 3, strong staining) and the percentage of positive tumor cells (score 0, none; score 1, 1–29%; score 2, 30–69%; score 3, >70%). Multiplying the staining intensity score by the positive tumor cell score generated the overall immunohistochemistry score. The final score ranged from 0 to 9 and was interpreted as follows: negative (0), weak (1–3), moderate (4–6), and strong (>6). NOX4 expression was classified as high (grade 4–9) and low (grade 0–3).

### 2.6. Cell Culture

HepG2 (hepatocellular carcinoma; Cat. No. CL-0103), Huh7 (hepatocellular carcinoma; Cat. No. CL-0120), and hepa1-6 (hepatocellular carcinoma; Cat. No. CL-0103) were purchased from the Procell Life Science & Technology (Wuhan, China). All of the above cells were cultured in 10% fetal bovine serum-containing DMEM (Gibco, Grand Island, NE, USA) and maintained at 37 °C in a 5% CO_2_ atmosphere.

### 2.7. Plasmid Constructions

The forward primer, CTCGAG (XhoI restriction site) ATGAATGTCCTGCTTTTCT, and the reverse primer, CCCGGG (SmaI restriction site) TCAGCTGAAAGACTCTTTA, were used to amplify the NOX4 from DNA extracted from HCC tissue specimens. The PCR product was then inserted into the LV-19-puro vector to generate a NOX4 overexpression plasmid. ShRNA primer sequences directed against CDS of NOX4 mRNA were designed (the forward primer, 5′-CCGGCCCTCAACTTCTCAGTGAATTCTCGAG AATTCACTGAGAAGTTGAGGGTTTTTG-3′, and the reverse primer, 5′-AATTCAAAAA CCCTCAACTTCTCAGTGAATTCTCGAGAATTCACTGAGAAGTTGAGGG-3′). The primers were annealed and ligated to lentiviral vector plasmid pLKO.1 using Age1 and EcoR1 restriction endonucleases. All plasmids were sequenced to confirm correctness.

### 2.8. Lentivirus Packaging and Stable Cell Lines

After the NOX4 overexpression plasmid was co-transfected with pMD2G and psPAX2 at a ratio of 4/3/2 into 293T cells, Huh7 and HepG2 cells were transduced for 48 h in the environment after the cell culture media had been collected and filtered with a 0.45 um filter. Puromycin (Sangon, Shanghai, China, 2 μg/mL) was then added to select NOX4-OE cells and the LV19-puro empty vector was used to generate OE-CON cells. The NOX4 shRNA lentivirus method is similar. The NOX4-shKD plasmid and the packaging plasmid (pMD2G and psPAX2) were used to transfect into 293T cells according to the proportion for 48 h and hepatoma cells were transduced with shRNA lentivirus. Stable cell line screening was conducted with the addition of puromycin. Then, 2 days later, 4 μg/mL of puromycin was added to select positive cells for 14 days. After 24 h of transfection, clones were screened by incubation in a complete medium containing 2 μg/mL puromycin for 48 h and then transferred to a complete medium without puromycin for further incubation. Transfected cells are used for further experiments (at least three clones overexpressing or down-expressing the gene of interest are selected and each clone is subjected to experiments).

### 2.9. RNA Extraction and Quantitative Real-Time PCR

TRIzol reagent (Takara, Kyoto, Japan) was used to extract the total RNA and the PrimeScript RT-PCR kit (Takara, Japan) was used to reverse-transcribe 1 g of total RNA in accordance with the manufacturer’s instructions. Each individual gene’s expression was detected using RT-qPCR (Applied Biosystems, Grand Island, NY, USA) and the SYBR-Green technique (Takara, Japan). The relative NOX4 expression levels were measured using the 2Ct technique and normalized to 18S RNA. A 10 min initial denaturation at 95 °C was followed by 40 cycles of PCR cycling at 94 °C (30 s), 60 °C (30 s), and 72 °C. 

The primers were as follows:

NOX4-forward-5′-TGACGTTGCATGTTTCAGGAG-3′

NOX4-reverse-5′-AGCTGGTTCGGTTAAGACTGAT-3′

AURKA-forward-5′: GGAATATGCACCACTTGGAACA-3′

AURKA-reverse-5′: TAAGACAGGGCATTTGCCAAT-3′

CCNB1-forward-5′: AATAAGGCGAAGATCAACATGGC-3′

CCNB1-reverse-5′: TTTGTTACCAATGTCCCCAAGAG-3′

CCNB2-forward-5′: TGCTCTGCAAAATCGAGGACA-3′

CCNB2-reverse-5′: GCCAATCCACTAGGATGGCA-3′

CDC20-forward-5′: GACCACTCCTAGCAAACCTGG-3′

CDC20-reverse-5′: GGGCGTCTGGCTGTTTTCA-3′

NCAPG-forward-5′: ATCCAGAAGTTAGACGGGCAG-3′

NCAPG-reverse-5′: GTGCGCCCTACAATTTTTGGC-3′

TOP2A-forward-5′: TGGCTGTGGTATTGTAGAAAGC-3′

TOP2A-reverse-5′: TTGGCATCATCGAGTTTGGGA-3′

MAD2L1-forward-5′: ATCACAGCTACGGTGACATTTC-3′

MAD2L1-reverse-5′: GCGGACTTCCTCAGAATTGGT-3′

GAPDH-forward-5′-CTGGGCTACACTGAGCACC-3′

GAPDH-reverse-5′-AAGTGGTCGTTGAGGGCAATG-3′

The results were expressed relative to the number of GAPDH transcripts used as an internal control.

### 2.10. Western Blotting

Using protein extraction buffer (Solarbio, RIPA lysis buffer, Beijing, China), the protein was extracted from HCC tissues (tumor and matched normal adjacent tissues) and cell lines. Protein concentrations were determined via BCA Protein Assay Kits (Elabscience, Wuhan, China). On a 10% SDS-PAGE, protein sample fractions were separated and the remaining fractions were then transferred to nitrocellulose membranes (Biosharp, Shanghai, China). After being blocked overnight at 4 °C with 5% fat-free milk, primary antibodies were exposed to membranes. After that, secondary antibodies that were HRP-conjugated were applied to the membranes for an hour at room temperature. Mouse anti-β-actin (Proteintech, 66009-1-1g, WB 1:5000, Wuhan, China), mouse anti-NOX4 (Proteintech, 67681-1-1g, WB 1:2000), rabbit anti-E-cadherin, and rabbit anti-Snail (Cell Signaling Technology, both WB 1:1000, Boston, MA, USA) were used as the main antibodies. HRP-conjugated goat anti-rabbit antibodies were used as the secondary antibodies (Cell Signaling Technology, WB 1:5000, Boston, MA, USA).

### 2.11. Cell Proliferation, Colony Formation, Wound Healing, and Intracellular ROS Detection In Vitro

In total, 1 × 10^4^ Huh7 and HepG2 cells were transfected and then distributed into 96-well plates. Ten microliters of CCK-8 test reagent were added to the cell-containing plates and incubated at room temperature for an additional two hours. To assess cell growth, the absorbance at 450 nm was measured for each well. Drugs were tested on cells grown in vitro in plates. Sorafenib was formulated in DMSO with a gradient concentration of 0–30 μM and added one day after cell attachment. Cell proliferation was evaluated after two days of treatment.

After being transfected, Huh7 and HepG2 cells were planted in 6-well dishes with 500 cells each and 2.5 mL of media and they were kept alive for 20 days. Crystal violet was used to stain and count cell-proliferating colonies that contained at least 50 cells. The colonies were stained with 0.1% crystal violet for 5 min after being fixed with 20% methanol for 30 min. The colonies were then counted after they had been washed three times with PBS. Only the positive (>50 cells/colony) colonies on the dishes were counted and compared.

After being transfected, 5 × 10^5^ Huh7 and HepG2 cells were spread in 6-well plates. The non-adherent cells were removed with PBS once the cells reached confluence. Three scratch incisions were made on the cell monolayer using a pipette tip (10 μL) and the non-adherent cells were subsequently washed away with PBS. Fresh DMEM without FBS was then used as the new solution. The distance between the wound edges was recorded at 0 and 36 h.

DCFH-DA was used as the ROS indicator and flow cytometry was used to quantify the intracellular ROS. In 6-well plates, NOX4-OE and OE-CON HepG2/Huh7 cells were plated and grown for 24 h. A DCFH-DA solution (10 μM) was applied after washing the cells with PBS and the cells were then incubated with the solution for 15 min. The cells were gathered and subjected to a flow cytometric analysis.

### 2.12. Nude Mouse Tumor Growth Analysis

Male nude mice aged 5 weeks were purchased from the Central South University’s Laboratory Animal (Changsha, China). Over the two weeks prior to the trial, the mice acclimated to a 12-h light–dark cycle with 50% humidity and free access to food and drink. Nude mice (n = 10) were injected subcutaneously with an approximate aliquot of 0.1 mL of the NOX4 overexpressed and down-regulated Huh7/HepG2 cells, which had been resuspended at a concentration of 5 × 10^7^ cells/mL. Two or three times a week, tumor volume was determined by multiplying the greater diameter by the smaller diameter. Then, 30 days later, the mice were killed and the tumor transplants were taken out for analysis. The Institutional Animal Care and Use Committee of the Xiangya School of Medicine gave its approval for these animal experiments.

### 2.13. Statistical Analysis

The statistical analysis was carried out using GraphPad Prism and SPSS software (version 23.0; SPSS Inc., Chicago, IL, USA). All experimental results are presented as the mean standard deviation of data from at least three separate replication trials. Our study was a cross-sectional study with a dichotomous variable in the type of outcome indicator and the sample size was calculated based on sensitivity and specificity. In this example, we have a non-inferiority design with a test level of a = 0.05 and a power of 1-β = 0.9, with parallel groups of 1:1. We used the logrank method in the PASS software to calculate summary statements from the PASS software revealed that a two-sided logrank test with a total sample size of 137 subjects (68 in the control group and 69 in the treatment group) has 90.2% power at a significance level of 0.050 to identify a hazard ratio of 0.5000 when the hazard rate for the control group is 1.00. Considering the lack of follow-up, therefore, a total of 150 subjects were included in this study. To compare the variations between the groups, the *t*-test was employed. The Fisher’s exact and chi-squared tests were used to investigate correlations between NOX4 and clinicopathological variables. In order to evaluate the overall survival (OS) and disease-free survival, Kaplan–Meier and logrank tests were used (DFS). Both univariate and multivariate analyses, using the Cox proportional hazards model, were utilized to identify the prognostic factors. The threshold for statistical significance was set at P 0.05. R software (version 4.02) was used for all data processing.

## 3. Results

### 3.1. Differentially Expressed Genes (DEGs) Analysis in HCC

To find the DEGs in HCC, the datasets of the associated clinical data were downloaded from GEO and Xena (https://xenabrowser.net/datapages/, accessed on 1 August 2022). This study comprised 304 non-tumor tissues and 364 cases of HCC altogether. Figure 2A’s cluster analysis graph shows that 412 genes were down-regulated and 196 genes were up-regulated when compared to normal tissues.

We used LASSO modeling on the TCGA expression matrix and prognostic data to find the main genes influencing the development of HCC and chose eight genes for risk score computation. (Figure 2B). The cut-off number was chosen using the rocr program and the patients were split into two groups based on their risk scores: high and low. An HR of 3.27 (1.91 to 5.6) was established for the group with elevated risk scores (Figure 2C). We built a nomogram for presentation and used calibration graphs to evaluate the nomogram’s predictive power in order to numerically assess the model’s survival forecasting ability. R^2^ = 0.75 indicates that our model has a strong predictive ability based on the model’s forecast (Figure 2D). Patients with low risk had a better chance of surviving than those with high risk, according to the K–M survival curve (Figure 2E, *p* < 0.001). There was a correlation between this risk score and the OS of patients, as shown in Figure 2F, which showed that the fraction of survivors declines as the risk score rises.

### 3.2. Hub Gene Identification and Functional Enrichment Analysis

The immune system is crucial to the different ways HCC patients are treated. Immune negative-regulated molecule up-regulation can prevent lymphocytes from destroying tumor cells, encourage tumor escape, and increase drug tolerance. We looked at how the eight key genes and immune checkpoints in the TCGA cohort correlated; the results revealed that NOX4 expression is positively associated with the immunosuppressive molecule SIGLEC7 and negatively correlated with NK cells (Figure 3A,B). The next step was to create qPCR primers for the eight chosen genes: CDC20, NCAPG, CCNB1, MAD2L1, AURKA, TOP2A, NOX4, and CCNB2. Following the collection of 20 patients’ fresh HCC specimens for confirmation, we found that NOX4 was much more highly expressed in the tumor tissues than in paracancerous tissues while the expression of other genes differed inconsistently in cancer and the paracancerous tissues of different patients (Figure 3C). T cells play a major role in killing tumors in the tumor microenvironment and our GSEA enrichment analysis showed two significant signaling pathways including T cell-mediated immunity (Appendix A) and positive regulation of cell killing (Appendix A). Additionally, we investigated the relationship between NOX4 and the anticipated ICB reaction signatures. The enrichment values for all immunotherapy-related positive signatures had a negative correlation with NOX4 (Figure 3D). NOX4-high expression and NOX4-low expression HCC microenvironmental immune cell analysis is depicted in Appendix A. Finally, we investigated the relationship between NOX4 expression in the TCGA dataset and the overall survival (OS) and disease-free survival (DFS) of patients. According to our study, patients with HCC with high NOX4 expression had worse OS and DFS rates than those with low NOX4 expression (Figure 3E,F).

### 3.3. Association of NOX4 Expression with Clinical Characteristics and Prognosis of HCC Patients

To further investigate the expression and clinical importance of NOX4 in HCC, immunohistochemical staining was performed on 150 paraffin-embedded HCC tissues from our hospital. We discovered that HCC tissues had considerably higher NOX4 expression than the surrounding healthy liver tissues (Figure 4A). The Western blot confirmation of NOX4 expression in 20 pairs of HCC specimens (tumor and matching nontumor tissues) from our institution was shown in Figure 4B. As anticipated, HCC tissues showed much higher levels of NOX4 protein expression than their non-tumor counterparts. HCC cells primarily showed positive NOX4 staining in their cytoplasm and nucleus. In total, 86 (57.3%) of the 150 HCC tissue samples exhibited high NOX4 expression levels (NOX4++ or NOX4+++) while 64 (42.7%) of the samples had low NOX4 expression (NOX4 or NOX4+) (Appendix A). These findings demonstrated that NOX4 expression levels varied between samples and were often greater in HCC tissues.

Table 1 provided an overview of the clinicopathological analysis findings which showed that the NOX4 expression did not correlate with age, the number of tumors, Child–Pugh class, etiologies of hepatitis, cirrhosis, α-fetoprotein (AFP) levels, vascular invasion, or differentiation grade. Kaplan–Meier survival analyses showed that high NOX4 expression levels were linked to significantly poorer OS (Figure 4C) and DFS (Figure 4D). Cox proportional hazards regression analysis indicated that NOX4 expression (*p* < 0.001), age (*p* = 0.026), sex (*p* = 0.036), number of tumors (*p* = 0.001), vascular invasion (*p* = 0.006), median differentiation grade (*p* = 0.038), and low differentiation grade (*p* = 0.001) were significantly associated with OS in HCC (Table 2). In addition, multivariate analysis revealed that NOX4 expression (*p* < 0.001), age (*p* = 0.039), number of tumors (*p* = 0.002), and low differentiation grade (*p* = 0.002) were independent prognostic factors for OS in HCC (Table 2). To quantitatively evaluate the predictive value of NOX4 expression for survival, we constructed a nomogram for display and used calibration plots to assess the predictive ability of the nomogram (Appendix A). These findings imply that NOX4 expression is an important predictor of HCC’s malignant development.

### 3.4. NOX4 Knockdown Decreased the Proliferation and Migration of HCC Cells In Vitro and In Vivo

Furthermore, to generate an HCC cell line with stable NOX4 knockdown (NOX4-KD), Huh7 and HepG2 cells were infected with lentiviruses designed to express NOX4 shRNA that targets NOX43′-UTR, followed by puromycin selection. The efficiency of the NOX4 knockdown in these cells was validated by Western blot analysis (Figure 5A,C). Consistently, we discovered that NOX4 knockdown significantly reduced proliferation (Figure 5B,D), colony formation (Figure 5E,F), and tumor proliferation in mice compared to the control group (Figure 5G–I). Taken together, these results support the hypothesis that NOX4 promotes tumor growth in HCC.

### 3.5. Overexpression of NOX4 Promoted HCC Cell Proliferation In Vitro and Vivo

We further stably overexpressed the endogenous NOX4 in Huh7 and HepG2 cells using a lentivirus; a Western blot was used to confirm NOX4 expression in the infected cells (Figure 6A,C). Then, we performed cell proliferation and colony formation assays to examine the function of NOX4 in HCC. The up-regulation of NOX4 was found to support the proliferation of Huh7 and HepG2 cells using the CCK-8 assay (Figure 6B,D). Colony formation assay results also support that NOX4 overexpression increased the proliferation of Huh7 and HepG2 cells (Figure 6E,F). To further assess NOX4’s impact on tumors, we performed a xenograft development experiment in nude mice employing both Huh7 cells overexpressing NOX4. Tumors from the NOX4-overexpressing group formed more rapidly and had significantly higher tumor volumes and weights when compared to the control group (Figure 6G–I).

### 3.6. NOX4 Influenced HCC Cell Migration: The Epithelial–Mesenchymal Transition (EMT)

The GSEA enrichment analysis was carried out with the aid of the online database to conduct a preliminary study of the biological function of the DEGs. The two signaling pathways where the elevated DEGs were most concentrated were the ECM–receptor interaction pathway and the negative regulation of cell adhesion signaling (Figure 7A–C). Since EMT is believed to be essential for the invasion and spread of cancer and to be a key component of the majority of malignancies, in Huh7-NOX4-KD cells we looked at the expression of epithelial and mesenchymal markers as well as other molecules thought to be involved in EMT. Figure 7D illustrated the results of a Western blot examination of Huh7 cells transfected with shRNA which revealed enhanced expression of E-cadherin and decreased expression of mesenchymal SNAI1 biomarkers. The wound healing assay revealed that downregulating NOX4 reduced the ability of Huh7 and HepG2 cells to migrate (Figure 7E), while NOX4-overexpressing Huh7 and HepG2 cells improved wound healing (Figure 7F). These results collectively provide compelling evidence that NOX4 encourages HCC invasiveness through EMT.

### 3.7. NOX4 Elevation Is Associated with Therapeutic Resistance and Immunotherapy Tolerance

Recent research indicated that EMT contributes to cancer chemoresistance and can be inhibited to overcome this resistance. Additionally, sorafenib resistance in HCC is said to be influenced by EMT. After sorafenib-resistant patients were sequenced, we discovered that their NOX4 expression was considerably higher than that of non-resistant patients (Figure 8A). In order to estimate the IC50 in two groups of HCC patients with high and low NOX4 expression, we used the pRRophetic algorithm to build a Ridge regression model based on the GDSC database expression profile and the TCGA gene expression profile. Hence, it was speculated that NOX4 may play an important role in sorafenib resistance (Figure 8B) as well as Lenalidomide, Gefitinib, and Lapatinib (Appendix A). To investigate the role of NOX4 in the sensitivity of HCC to sorafenib, we generated HepG2 cells down-expressing NOX4. For HepG2-KD-CON and HepG2-KD cells, the sorafenib IC50 values were 15.34 μM and 8.87 μM, respectively (Figure 8C). These findings collectively imply that NOX4 overexpression causes relative sorafenib resistance in HCC (Appendix A). Previous research has shown that NOX4, related to respiratory metabolism, affects the effectiveness of medications by producing high amounts of ROS. Our experiments also found that ROS is significantly increased in NOX4 highly expressed Huh7 cells and HepG2 cells (Figure 8D and Appendix A).

Targeted therapy alone has not yet proven to be effective enough to treat HCC, so several clinical trials combining targeted therapy and immunotherapy have been carried out. In order to explore the efficacy of NOX4 and immunotherapy, we used the TIDE database to calculate the immune cell infiltration between NOX4 low expression and NOX4 high expression groups. In the statistically significant NOX4 high expression group, the proportion of macrophages was raised while the proportion of CD8^+^T cells was increased in the low NOX4 expression group (Figure 8E). Additionally, we examined the relationship between NOX4 expression and the effectiveness of immunotherapy in HCC using the Easlier package and discovered that the low NOX4 expression group had a greater percentage of immunotherapy response. (Figure 8F). Our research concluded that NOX4 knockdown could prevent EMT in HCC cells, boost antitumor immune effects, and overcome sorafenib resistance for tumor inhibition. (Figure 8G).

## 4. Discussion

HCC, which accounts for more than 90% of early malignant liver tumors and is distinguished by a high degree of malignancy, has a high rate of recurrence and a poor prognosis [10]. Therefore, the need for trustworthy biomarkers to categorize these patient groups at elevated risk is imperative. Simple methods for evaluating gene expression data, such as DEGs analysis, are useful to help identify important genes associated with HCC prognosis that we used in our study. First, we used the GEO and Xena databases to identify 608 DEGs between HCC and normal tissues. Afterward, we performed LASSO modelling on the TCGA expression matrix and selected eight genes (CDC20, NCAPG, CCNB1, MAD2L1, AURKA, TOP2A, NOX4, and CCNB2) for risk score calculation. We then developed a prognostic risk model based on the eight genes. Through the validation of several datasets, we found that this predictive risk model has excellent precision and sensitivity in forecasting the outcome of HCC patients. The other study used differential gene expression analysis to filter DEGs and identify hub genes. It then carried out additional analysis to investigate the association between the hub genes and diseases. The RT-PCR and Western blotting were used to confirm the hub genes’ expressed levels. Our study’s process was comparable to these works. [11,12]. However, there are several distinctions that should be noted. Firstly, diverse platforms’ data sources were selected for this study to make them more representative. Second, the significance of hub genes in our study was further strengthened by immune infiltration analysis and several validation trials.

The majority of the eight genes in our predictive model have been implicated in cancer according to reports. The cell cycle regulator protein CDC20 (cell division cycle 20 homolog), a major oncogenic component required for the mitosis process, is strongly expressed in many tumors [13,14]. A prognostic indicator for hepatocellular carcinoma called NCAPG has been linked to vascular invasion [15]. CCNB1 is an essential regulator of the cell cycle. The outcomes of this investigation were in agreement with those of our study. Through extensive bioinformatics research, High MAD2L1 expression in HCC was found to be substantially correlated with overall patient survival and clinical features by Qian Chen et al. [16]. In the prognostic analysis of HCC by other scholars, TOP2A is also an important predictor of HCC prognosis but the function of TOP2A is not fully understood [17]. Understanding the function of these genes may prompt the development of HCC treatments.

The expression of NOX4 mRNA was found to be significantly higher in HCC tissue when compared to non-cancerous liver tissue when we gathered fresh HCC specimens from 20 patients for the qPCR validation of 8 genes. NOX4 functions as a main oxygen sensor by transferring electrons from cytosolic NADPH across biological membranes to molecular oxygen and creating ROS [18]. Recent research has shown that a variety of physiological and pathological processes depend on intracellular signals from ROS. ROS levels are consistently elevated in tumors and some research suggests that ROS may have human carcinogenic effects [19,20]. The NADPH oxidase (NOX) family, which comprises seven members (NOX1–5, and DUOX1-2), is significant in the onset and development of cancer and has drawn growing attention as a significant source of ROS [6]. Among the NOX family members, NOX4 is the most often overexpressed NOX isoform associated with cancer; it has been observed in a number of solid tumors, including colorectal cancer (CRC) [21], pancreatic cancer [22], glioblastoma [23], and prostate cancer [24]. For instance, oleic acid (OA)-induced CRC cell metastasis requires the production of NOX4 followed by a rise in ROS levels [25]. NOX4 also links pancreatic cancer metabolic regulation to endoplasmic reticulum redox vulnerability [26]. In conclusion, NOX4 might have various functions in various cancers. However, the effect of NOX4 on HCC development and prognosis remains controversial. According to some research, high nuclear NADPH oxidase 4 expression is linked to cancer development and a bad prognosis in HCC [27]. In contrast, another study discovered that HCC patients with high NOX4 expression levels had improved overall survival [28]. In this study, we retrospectively analyzed 150 HCC specimens from our hospital and confirmed that NOX4 expression was frequently upregulated in HCC patients by IHC and that high NOX4 expression was associated with aggressive clinical features and poor prognosis. Our research showed that NOX4 was a poor independent prognostic variable through Cox proportional hazard regression analysis. Functionally, NOX4 knockdown inhibits HCC cell proliferation and tumor development in xenograft animal models while NOX4 overexpression promotes tumor progression, suggesting that NOX4 plays an oncogenic role in the growth of HCC.

We also performed a number of bioinformatics studies. By using GSEA enrichment analysis, it was discovered that two important signaling pathways—“ECM-receptor interaction” and “negative regulation of cell adhesion”—were significantly favorably linked to NOX4 expression. It has been noted that the interplay between ECM receptors and the cell adhesion pathway is crucial for the metastasis of cancer [29]. EMT, a biological process in which epithelial cells are changed into cells with the mesenchymal phenotype by a particular procedure, is one type of cancer metastasis [30]. Next, using wound healing tests, we found that NOX4 knockdown reduced the ability of Huh7 and HepG2 cells to migrate. Additional studies revealed that NOX4 expression inhibition decreased the mesenchymal marker snail expression while increasing the expression of the epithelial marker E-cadherin. These results showed that NOX4 may promote tumor growth by inducing EMT in HCC cells. It is crucial to note that many studies have shown that EMT is a common cause of sorafenib resistance in HCC and poor disease-free survival [31,32] and that snails can directly increase the ABC transporter ABCB1 in cancer cells, leading to treatment resistance [33]. Sorafenib is currently approved for the treatment of HCC but early findings suggest that the efficacy of sorafenib varies. Many HCC patients experienced tumor progression after 4–5 months of sorafenib treatment and eventually developed sorafenib resistance. Therefore, elucidating the causes of sorafenib resistance in HCC treatment and improving its therapeutic efficacy are the hotspots of current research.

Recent studies have shown that mechanisms of acquired sorafenib resistance include crosstalk in the PI3K/Akt and JAK-STAT pathways, activation of hypoxia-induced pathways, and the epithelial–mesenchymal transition [34]. Based on data in the literature, patients who fail sorafenib treatment may be suitable for second-line therapy or combination therapy [35,36,37]. In the current study, we demonstrated a correlation between elevated NOX4 levels and poor sorafenib responsiveness in HCC cells. Clinical evidence also revealed that sorafenib resistance was connected to increased NOX4 expression in HCC patients. This study has some limitations. First, future studies should concentrate on the molecular process of NOX4. Second, this was a single-center retrospective study, results may unavoidably be impacted by selection bias.

## 5. Conclusions

The salient and novel findings of this article are as follows. (a) A genetic model with clinical and genomic information from TCGA and GEO was developed; (b) NOX4 expression in HCC is higher than peritumor tissue and related to poor prognosis; and (c) the knockdown of NOX4 suppresses EMT in HCC cells which may reverse sorafenib resistance. In conclusion, our study suggests that targeting NOX4 would be an optimal therapeutic strategy in HCC and that NOX4 may function as a biomarker to predict the response to sorafenib therapy.

## Figures and Tables

**Figure 1 biomedicines-11-02196-f001:**
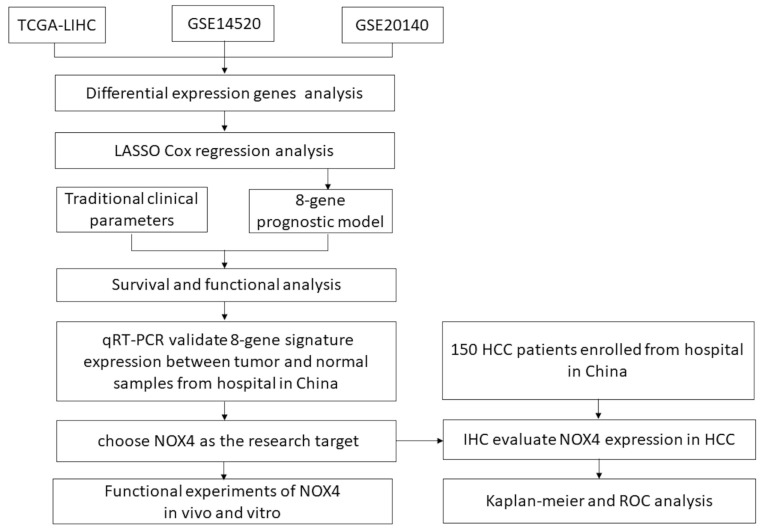
Flow diagram of the study.

**Figure 2 biomedicines-11-02196-f002:**
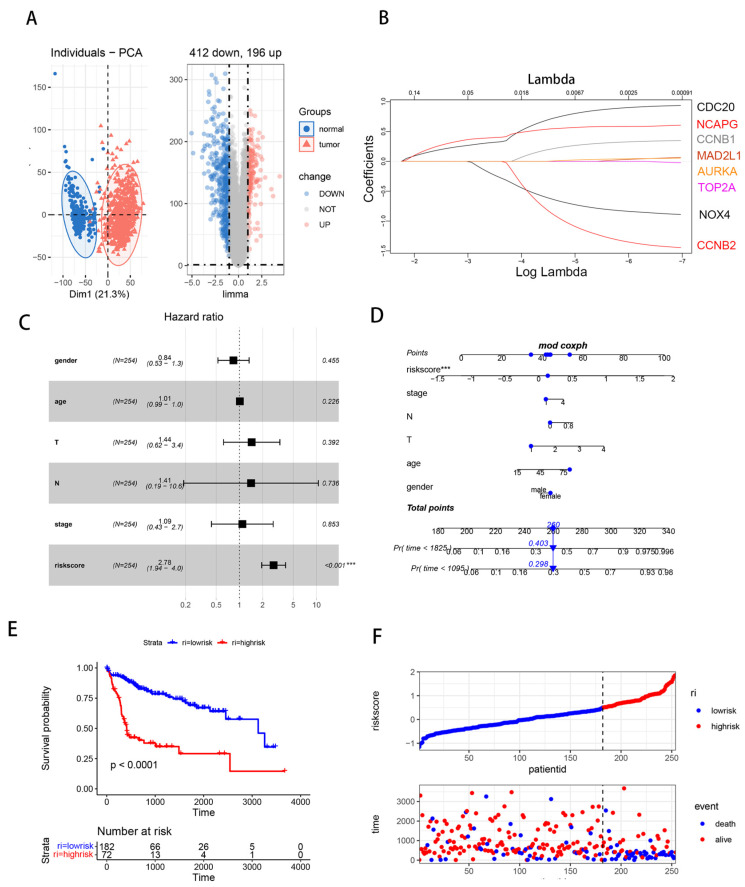
Identification of the candidate survival-related genes between HCC and normal tissues. (**A**) Differently expressed genes in the HCC group and normal group from the GEO datasets (TCGA, GSE14520, and GSE 20140) were presented in the PCA and volcano map. Red dots in the volcano map represent significant overexpression genes and blue dots represent significantly reduced genes. (**B**) Selecting the best λ in the LASSO model, the LASSO coefficient spectrum of 8 genes was enrolled to generate a coefficient distribution map. (**C**) Multivariate Cox regression analysis of the relationship between the risk score and clinicopathological characteristics regarding OS. (**D**) Nomogram constructed combined with risk score, TNM stage, age, and sex. (**E**) Survival analysis of 8 hub genes’ risk score in HCC based on the TCGA data. (**F**) Hub gene expression distribution and survival status based on TCGA. Data are presented as mean ± SD and are representative of three independent experiments. (*** *p* < 0.001).

**Figure 3 biomedicines-11-02196-f003:**
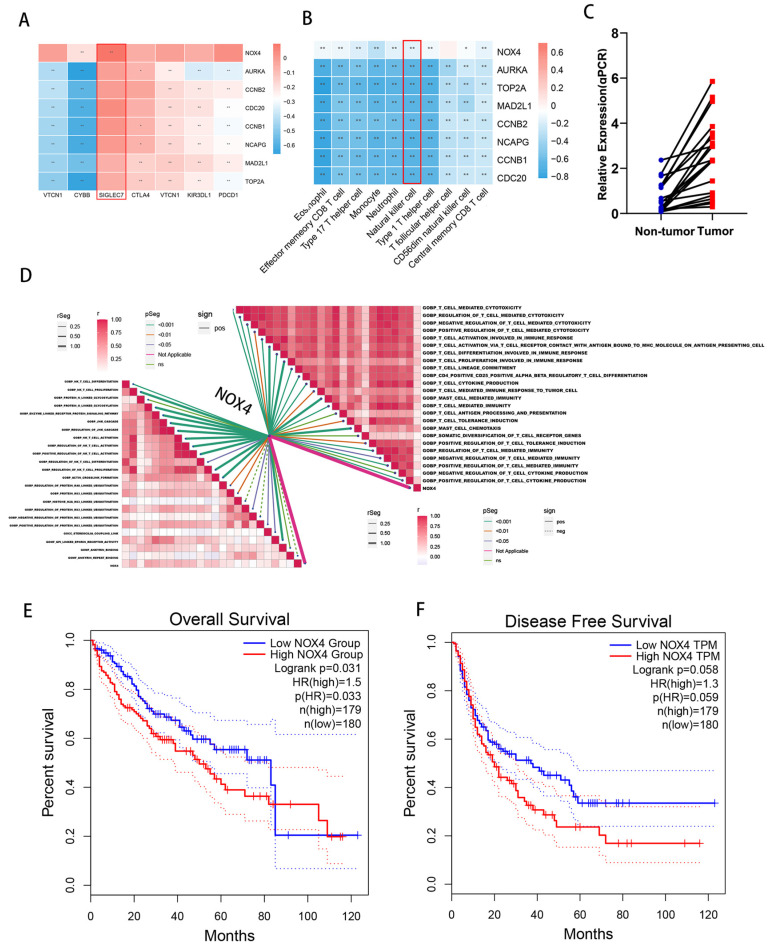
Screening for interest gene-NOX4 and functional enrichment analysis. (**A**) The correlation between the eight key genes and immune checkpoint. (**B**) The correlation between every key gene and each TME immune cell subtype. Blue, negative; Red, positive. (**C**) qPCR analysis of NOX4 in 20 paired HCC tissues (T) and matched adjacent noncancerous tissues (NT). The average NOX4 mRNA expression was normalized to the expression of GAPDH. (**D**) Correlations between NOX4 and the enrichment scores of T cell and NK cells-predicted pathways. (**E**,**F**) The correlation of NOX4 expression with patients’ overall survival and disease-free survival was conducted in the TCGA dataset, respectively. Analysis was performed on 1 August 2022 from the database GEPIA at http://gepia.cancer-pku.cn/. (* *p* < 0.05, ** *p* < 0.01).

**Figure 4 biomedicines-11-02196-f004:**
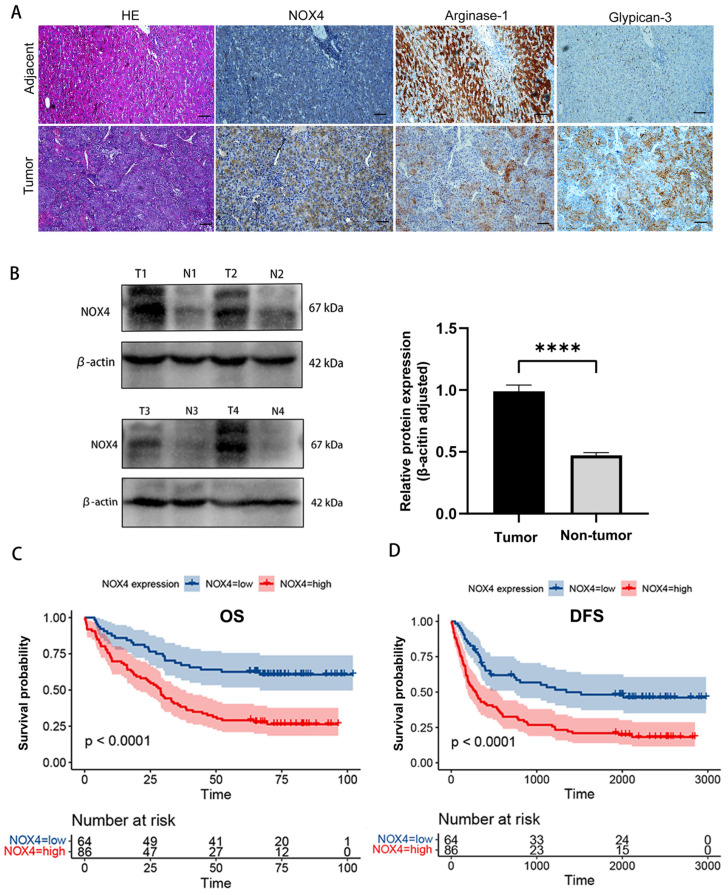
Upregulation of NOX4 expression in HCC is associated with unfavorable prognosis. (**A**) The typical IHC picture of NOX4, arginase-1, and glypican-3 in paired HCC and corresponding noncancerous liver tissues from the Xiangya cohort. (**B**) Western blotting (WB) analysis of NOX4 in 20 paired T and their NT. Levels of NOX4 quantified by optical density and normalized to β-actin levels in the same samples are shown on the left of WB results relative to levels in NT. (**C**) Overall survival analysis of 150 HCC patients with different NOX4 protein expression in the Xiangya cohort. (Logrank test: *p* < 0.001). (**D**) Disease-free survival analysis of 150 HCC patients with different NOX4 protein expression levels in the Xiangya cohort. (Logrank test: *p* < 0.001). Scale bar, 100 μm. (**** *p* < 0.0001).

**Figure 5 biomedicines-11-02196-f005:**
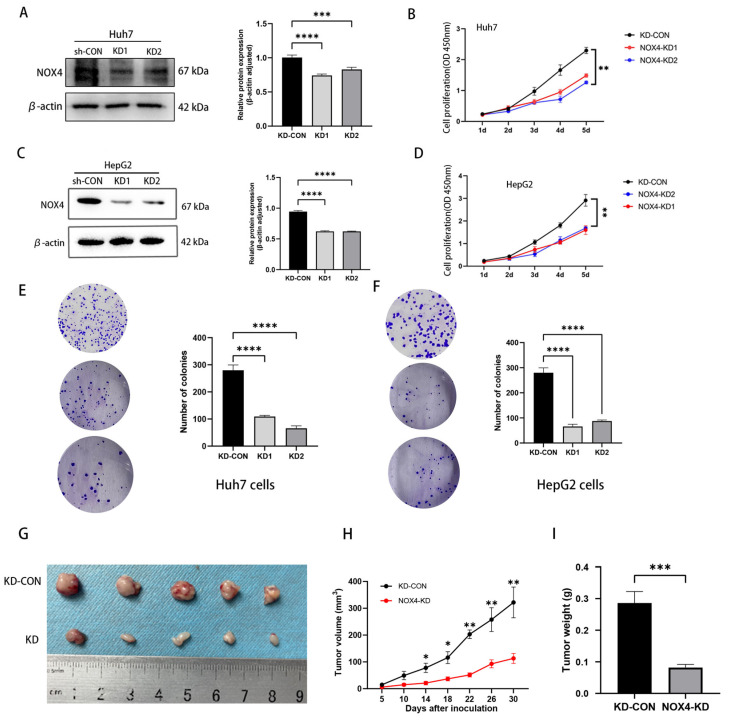
Knockdown of NOX4 inhibits HCC cell proliferation in vitro and xenograft tumor development in vivo. (**A**,**C**) WB analysis of Huh7 and HepG2 cells infected with NOX4-KD and negative control (KD-CON) lentivirus. Levels of NOX4 were quantified based on the optical density and were normalized to β-actin levels in the same samples. (**B**,**D**) CCK8 assays were performed to investigate the proliferation of Huh7 and HepG2 cells infected with NOX4-KD lentivirus. (**E**,**F**) Colony formation assays were performed to determine the colonies of cells with NOX4 knockdown. (**G**) NOX4 down-regulation inhibited the growth of HepG2 xenograft tumors (n = 5 in each group). These tumors were formed by injection of HepG2 cells with a NOX4 knockdown or carrying a negative control. (**H**) Tumor volumes were measured every 3 days. (**I**) Average tumor weights of the xenografts were recorded. (* *p* < 0.05, ** *p* < 0.01, *** *p* < 0.001 and **** *p* < 0.0001).

**Figure 6 biomedicines-11-02196-f006:**
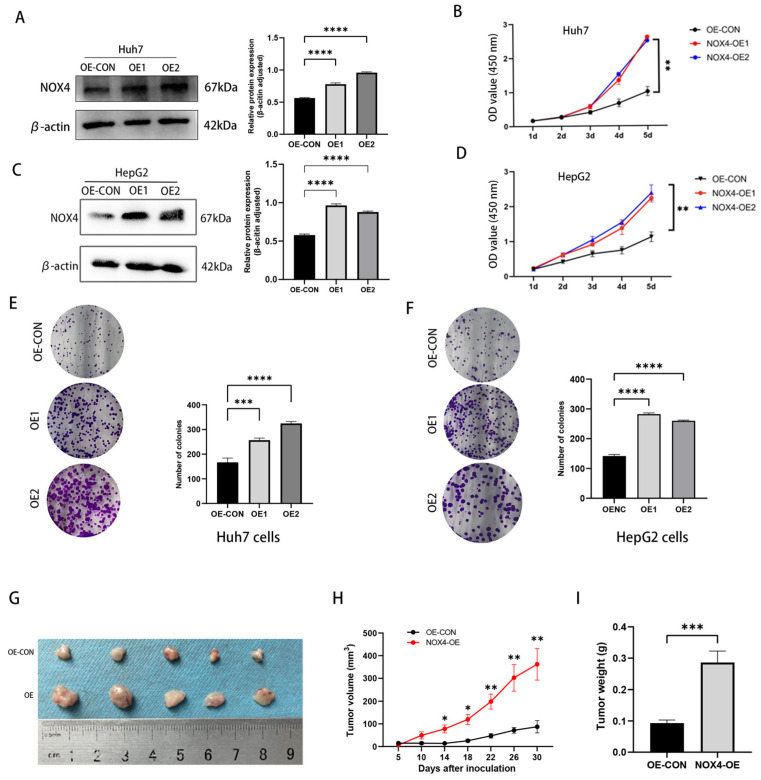
Overexpression of NOX4 promotes HCC cell proliferation in vitro and xenograft tumor development in vivo. (**A**,**C**) WB analysis of Huh7 and HepG2 cells infected with NOX4-OE and negative control (OE-CON) lentivirus. Levels of NOX4 were quantified on the basis of optical density and were normalized to β-actin levels in the same samples. (**B**,**D**) CCK8 assays were performed to investigate the proliferation of Huh7 and HepG2 cells infected with NOX4-OE lentivirus. (**E**,**F**) Colony formation assays were performed to determine the colonies of cells with NOX4 overexpression. (**G**) NOX4 up-regulation promoted growth of HepG2 xenograft tumors (n = 5 in each group). These tumors were formed by injection of HepG2 cells overexpressing NOX4 or carrying a negative control. (**H**) Tumor volumes were measured every 3 days. (**I**) Average tumor weights of the xenografts were recorded. (* *p* < 0.05, ** *p* < 0.01, *** *p* < 0.001, and **** *p* < 0.0001).

**Figure 7 biomedicines-11-02196-f007:**
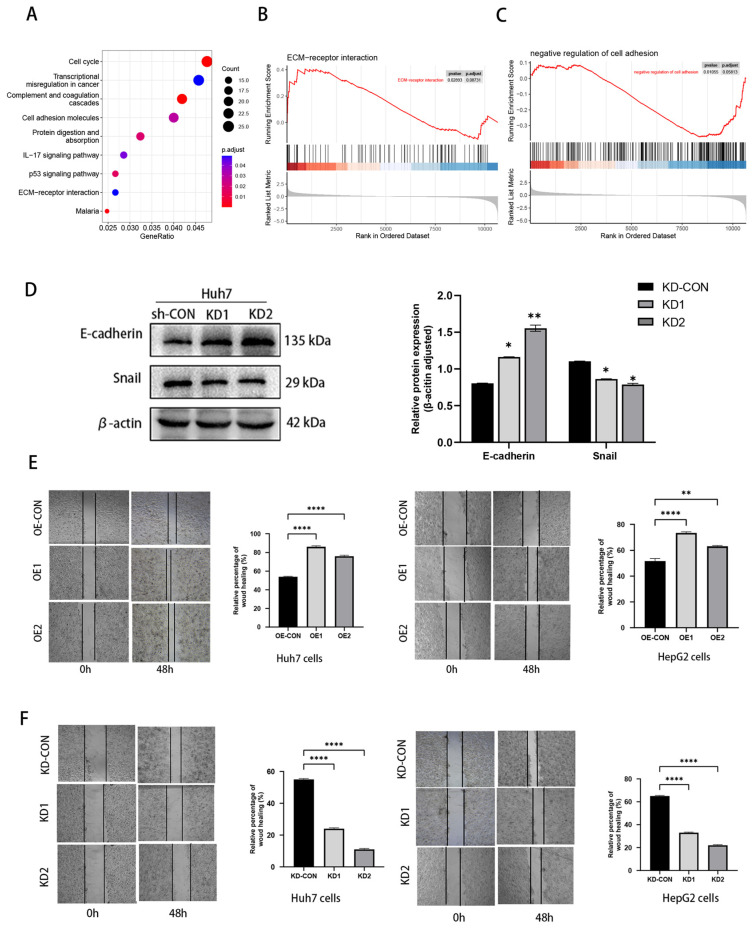
NOX4 promotes tumor migration in vitro by inducing EMT. (**A**) The bubble plot of the biological process enriched for the DEGs between the HCC group and normal group in the TCGA cohort and GSE14520 and GSE 20140 datasets. (**B**,**C**) The GSEA enrichment showed two significant signaling pathways including the ECM–receptor interaction pathway (**B**) and negative regulation of cell adhesion (**C**). (**D**) The protein levels of epithelial markers (E-cadherin) and mesenchymal markers (SNAI1) were measured by WB in Huh7 cells normalized to β-actin expression. (**E**,**F**) Wound healing assays were performed to examine the Huh7/HepG2 cell migration of NOX4 knockdown/overexpression. (* *p* < 0.05, ** *p* < 0.01, and **** *p* < 0.0001).

**Figure 8 biomedicines-11-02196-f008:**
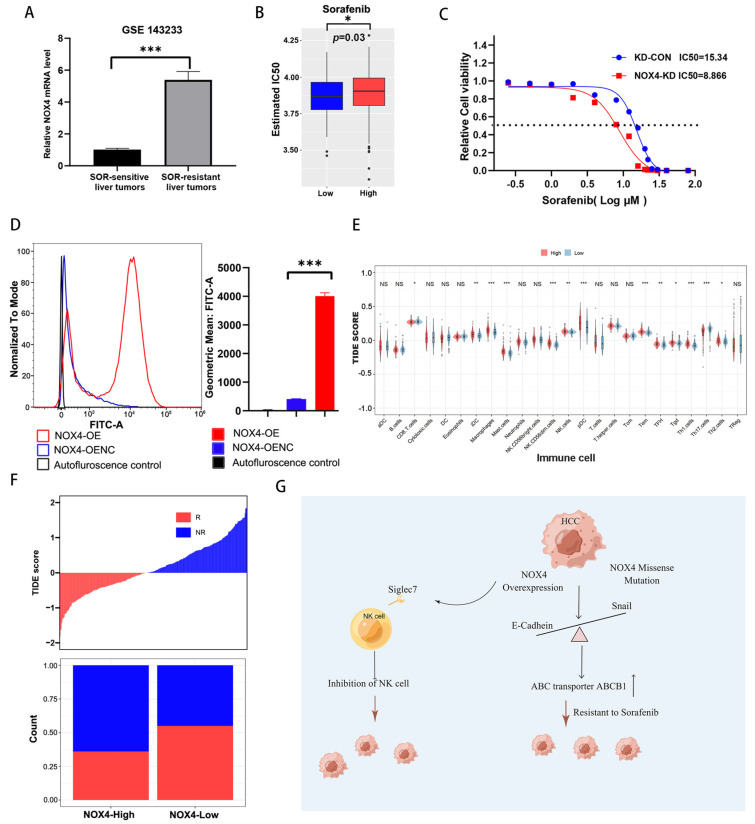
NOX4 is inversely correlated with HCC sensitivity to sorafenib. (**A**) The mRNA expression level of NOX4 in sorafenib-sensitive and sorafenib-resistant human liver tumors from GSE143233. (**B**) The box plots of the estimated IC50 for sorafenib are shown in NOX4-high and NOX4-low expression in LIHC in the TCGA cohort. (**C**) The IC50 of NOX4-KD and KD-CON in HepG2 cells after treatment with sorafenib for 72 h. (**D**) Representative histograms for the fluorescence induced by the DCFH-DA probe in NOX4-OE and OE-CON in Huh7 cells. (**E**) The TIDE database calculated the difference in proportion of immune cells between the two groups. (**F**) The EaSIeR package predicted the difference of immunotherapy cells between the two groups. (**G**) Working model. The molecular mechanisms of NOX4 promote HCC development and sorafenib resistance. Results shown are the mean ± SD of triplicate determinations from three independent experiments. (* *p* < 0.05, ** *p* < 0.01, *** *p* < 0.001, and NS No Significance).

**Table 1 biomedicines-11-02196-t001:** Association of NOX4 expression with the clinicopathological characteristics of HCC patients.

Characteristics	NOX4 Expression			
	Overall (n = 150)	Low (n = 64)	High (n = 86)	*p*-Value
Age (years, mean ± SD)	51.2 ± 12.1	48.9 ± 13.8	52.9 ± 10.3	0.057
Sex (n, %)				
male	130 (86.7)	51 (79.7)	79 (91.9)	
female	20 (13.3)	13 (20.3)	7 (8.1)	0.030
Number of tumors (n, %)				
single	131 (87.3)	58 (90.6)	73 (84.9)	
multiple	19 (12.7)	6 (9.4)	13 (15.1)	0.296
Child-Pugh class (n, %)				
A	134 (89.3)	58 (90.6)	76 (88.4)	
B	10 (6.7)	4 (6.3)	6 (6.9)	
C	6 (4.0)	2 (3.1)	4 (4.7)	0.877
Etiologies of hepatitis (n, %)				
none	20 (13.3)	6 (9.4)	14 (16.3)	
HBV	114 (76.0)	53 (82.8)	61 (70.9)	
HCV	5 (3.3)	1(1.6)	4 (4.7)	
alcohol	3 (2.0)	2 (3.1)	1 (1.2)	
others	8 (5.4)	2 (3.1)	6 (6.9)	0.312
Cirrhosis (n, %)				
absence	69 (46.0)	35 (54.7)	34 (39.5)	
presence	81 (54.0)	29 (45.3)	52 (60.5)	0.066
AFP (n, %)				
≤400 ng/mL	120 (80.0)	51 (79.7)	69 (80.2)	
>400 ng/mL	30(20.0)	13 (20.3)	17 (19.8)	0.934
Vascular invasion				
absence	130 (86.7)	59 (92.2)	71 (82.6)	
presence	20 (13.3)	5 (7.8)	15 (17.4)	0.086
Differentiation grade (n, %)				
high	18 (12.0)	9 (14.1)	9 (10.5)	
median	69 (46.0)	32 (50.0)	37 (43.0)	
low	63 (42.0)	23 (35.9)	40 (46.5)	0.415

Abbreviations: NOX4, NADPH oxidase 4; HCC, hepatocellular carcinoma; HBV, hepatitis B virus; HCV, hepatitis C virus; AFP, alpha-fetoprotein.

**Table 2 biomedicines-11-02196-t002:** Univariate and multivariate analysis of prognostic variables for overall survival in patients with HCC.

Prognostic Variables		Univariate Analysis		Multivariate Analysis	
	No. of Patients	HR (95% CI)	*p*	HR (95% CI)	*p*
Expression of NOX4					
low	64	[Reference]		[Reference]	
high	86	2.522 (1.588–4.005)	<0.001	2.397 (1.486–3.867)	<0.001
Age					
≤55 years	93	[Reference]		[Reference]	
>55 years	57	1.616 (1.059–2.465)	0.026	1.577 (1.024–2.429)	0.039
Sex					
female	20	[Reference]		[Reference]	
male	130	2.177 (1.050–4.510)	0.036	1.601 (0.758–3.382)	0.217
Number of tumors					
single	131	[Reference]		[Reference]	
multiple	19	2.542 (1.450–4.459)	0.001	2.533 (1.406–4.561)	0.002
HBV infection					
negative	36	[Reference]			
positive	114	1.332 (0.081–2.215)	0.270		
Cirrhosis					
absence	69	[Reference]			
presence	81	1.452 (0.945–2.231)	0.089		
AFP					
≤400 ng/mL	120	[Reference]			
>400 ng/mL	30	1.217 (0.724–2.0460	0.458		
Vascular invasion					
absence	130	[Reference]		[Reference]	
presence	20	2.103 (1.237–3.576)	0.006	1.155 (0.662–2.016)	0.611
Differentiation grade					
high	18	[Reference]		[Reference]	
median	69	2.994 (1.065–8.418)	0.038	2.485 (0.875–7.053)	0.087
low	63	6.150 (2.211–17.108)	0.001	5.310 (1.881–14.989)	0.002

Abbreviations: HCC, hepatocellular carcinoma; NOX4, NADPH oxidase 4; HBV, hepatitis B virus; AFP, alpha-fetoprotein.

## Data Availability

In this study, publicly available datasets were analyzed. The dataset in this article was obtained from the open source databases TCGA (http://cancergenome.nih.gov/, accessed on 1 August 2022); GSE14520 and GSE 20140 were obtained from the GEO database (http://www.ncbi.nlm.nih.gov/geo/, accessed on 1 August 2022).

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
