# Peer review of "Identification of NOX4 as a New Biomarker in Hepatocellular Carcinoma and Its Effect on Sorafenib Therapy"

_biomedicines, 2023, doi:10.3390/biomedicines11082196_

Round 1
Reviewer 1 Report
In this study, the authors by means GEO and TCGA dataset tried to explore the differential co-expressed genes and their prognostic correlation between HCC and normal samples and then results were validated by qRT-PCR in paired fresh HCC samples. An eight-gene model was effective in predicting the prognosis of HCC patients in the validation cohorts. Based on qRT-PCR results, NOX4 was selected to further explore biological functions within the model with 150 cases of paraffin-embedded HCC tissues further scored for NOX4 immunohistochemical staining. They found found that the NOX4 expression was significantly upregulated in HCC and was associated with poor survival. Of interest, knockdown of NOX4 markedly inhibited the progression of HCC in vivo and in vitro. The sensitivity of HCC cells to sorafenib therapy was decreased after NOX4 overexpression. They concluded that NOX4 overexpression is associated with poor prognosis of HCC and might serve as a therapeutic target for HCC and a biomarker for predicting response to sorafenib treatment.
The study is of interest since identification of biomarkers are of major clinical relevance for HCC patient prognosis prediction and therapeutic target identification and response prediction. However, some issues deserve further data and should be addressed.
-paraffin-embedded HCC samples: could the authors describe main clinical features of enrolled patients. In particular, viral vs non-viral underlying liver disease, HCC stage, alfa-fetoprotein serum levels.
-sorafenib resistance: in the discussion the authors stated that several clinical trials combining targeted therapy and immunotherapy have been carried out. This is a very important issue because the authors should recall important literature data on HCC therapies: 1) most recent and ongoing clinical trial on combination treatment based on tyrosine kinase inhibitors plus immune checkpoint inhibitors enrolled patients not previously treated and randomized with sorafenib arm, as well-described and summarized in a recent comprehensive review worth mentioning (TKIs in combination with immunotherapy for hepatocellular carcinoma. Expert Rev Anticancer Ther. 2023 Mar;23(3):279-291. doi: 10.1080/14737140.2023.2181162).
2) An interesting and unsolved issue is related to the good anti-tumor efficacy of regorafenib, an agent that structurally is very similar to sorafenib, assessed in the RESORCE trial and several real-word/clinical practice studies enrolling patients who progressed under sorafenib, as summarized in a comprehensive review (Experience with regorafenib in the treatment of hepatocellular carcinoma. Therap Adv Gastroenterol. 2021 May 28;14:17562848211016959. doi: 10.1177/17562848211016959.). A comment on potential reason explaining the good treatment response to regorafenib in sorafenib-resistant patients could improve the discussion.
- Of therapeutic interest, there are cohort studies demonstrating that HCC patients who progressed under first-line sorafenib, have shown in the second-line setting a good anti-tumor response to metronomic capecitabine treatment, likely to the different anti-tumor mechanism, as recently demonstrated (Metronomic capecitabine as second-line treatment in hepatocellular carcinoma after sorafenib failure. Dig Liver Dis. 2015 Jun;47(6):518-22; Metronomic capecitabine as second-line treatment for hepatocellular carcinoma after sorafenib discontinuation. J Cancer Res Clin Oncol. 2018 Feb;144(2):403-414.). As an anti-tumor agent of potential efficacy after sorafenib failure, these studies should be recalled.
-Introduction: to further improve the current HCC scenario and further highlight the importance of predictive markers for HCC management improvement, the authors should recall recently described epidemiological changing scenario of HCC due to a progressive increase of non-viral cases and, improved outcomes of ablative and transarterial treatments, an improved overall survival thanks to systemic treatment development, as recently demonstrated (The changing scenario of hepatocellular carcinoma in Italy: an update. Liver Int. 2021 Mar;41(3):585-597. doi: 10.1111/liv.14735.).
Author Response
Point 1: Paraffin-embedded HCC samples: could the authors describe main clinical features of enrolled patients. In particular, viral vs non-viral underlying liver disease, HCC stage, alfa-fetoprotein serum levels.
Response 1: The cases we collected were liver cancer patients from China, so most of them were liver cancer developed from hepatitis B virus infection and cirrhosis. The NOX4 expression did not correlate with age, the number of tumors, Child-Pugh class, etiologies of hepatitis, cirrhosis, α-fetoprotein (AFP) levels, vascular invasion, or differentiation grade.
Point 2: Sorafenib resistance: in the discussion the authors stated that several clinical trials combining targeted therapy and immunotherapy have been carried out. This is a very important issue because the authors should recall important literature data on HCC therapies: 1) most recent and ongoing clinical trial on combination treatment based on tyrosine kinase inhibitors plus immune checkpoint inhibitors enrolled patients not previously treated and randomized with sorafenib arm, as well-described and summarized in a recent comprehensive review worth mentioning (TKIs in combination with immunotherapy for hepatocellular carcinoma. Expert Rev Anticancer Ther. 2023 Mar;23(3):279-291. doi: 10.1080/14737140.2023.2181162).
Response 2: Important literature data on drug therapy for HCC have been added to the discussion section.
Point 3: An interesting and unsolved issue is related to the good anti-tumor efficacy of regorafenib, an agent that structurally is very similar to sorafenib, assessed in the RESORCE trial and several real-word/clinical practice studies enrolling patients who progressed under sorafenib, as summarized in a comprehensive review (Experience with regorafenib in the treatment of hepatocellular carcinoma. Therap Adv Gastroenterol. 2021 May 28;14:17562848211016959. doi: 10.1177/17562848211016959.). A comment on potential reason explaining the good treatment response to regorafenib in sorafenib-resistant patients could improve the discussion.
Response 3: Potential reasons why sorafenib resistant patients respond well to rigorafenib therapy have been discussed in the discussion section.
Point 4: Of therapeutic interest, there are cohort studies demonstrating that HCC patients who progressed under first-line sorafenib, have shown in the second-line setting a good anti-tumor response to metronomic capecitabine treatment, likely to the different anti-tumor mechanism, as recently demonstrated (Metronomic capecitabine as second-line treatment in hepatocellular carcinoma after sorafenib failure. Dig Liver Dis. 2015 Jun;47(6):518-22; Metronomic capecitabine as second-line treatment for hepatocellular carcinoma after sorafenib discontinuation. J Cancer Res Clin Oncol. 2018 Feb;144(2):403-414.). As an anti-tumor agent of potential efficacy after sorafenib failure, these studies should be recalled.
Response 4: Recent advances in targeted drug therapy for HCC have been added to the discussion sections as recommended.
Point 5: Introduction: to further improve the current HCC scenario and further highlight the importance of predictive markers for HCC management improvement, the authors should recall recently described epidemiological changing scenario of HCC due to a progressive increase of non-viral cases and, improved outcomes of ablative and transarterial treatments, an improved overall survival thanks to systemic treatment development, as recently demonstrated (The changing scenario of hepatocellular carcinoma in Italy: an update. Liver Int. 2021 Mar;41(3):585-597. doi: 10.1111/liv.14735.)
Response 5: Recent global changes in the epidemiology of HCC have been discussed in the introduction section as requested.
Reviewer 2 Report
The manuscript entitled “NOX4: a potential therapeutic target for hepatocellular carcinoma and its mechanism underlying sorafenib resistance” focuses on study of NOX4 expression in HCC patients and its correlation with the cancer cell proliferation and migration as well as epithelial-mesenchymal transition and the cancer patients’ survival. The manuscript could be of interest to the journal audience. However, some issues should be addressed before publication:
1. Title is suitable for review article, but not for research article. It is recommended to change it so to reflect the main results obtained in this study.
2. In the Abstract, the authors stated “Effectively biomarkers were necessary for HCC patient prognosis prediction and therapeutic target identification”. This sentence is confusing because it is necessary to distinguish the terms “prognosis” and “prediction”. Also, this difference should be discussed in the Introduction. Use the following papers: doi: 10.1016/j.ejca.2008.03.006 and doi: 10.1080/14737159.2021.1987217.
3. The Introduction section is very poor, there is no discussion on roles NOX4 as NADPH oxidase and its implication in cancer. So, it is recommended to shortly discuss this using the following papers: doi: 10.3389/fcell.2022.884412; doi: 10.3389/fcell.2022.884412, etc.
4. Results section: in the second subsection the authors described the identification of hub genes, however, before this, Fig. 2 depicts relationships between clinicopathological characteristics and the hub genes – the logical presentation is disturbed.
5. Fig. 2C – is there any correlation between DEGs and clinicopathological characteristics? Fig. 2E – are there survival rate for HCC patients or DEGs or hub genes?
6. Also, it is not clear, why on lines 306-317, the authors discussed the immune system and immune checkpoints – which of identified genes do regulate them? Also, on lines 470-472, results on the association of NOX4 expression and levels of macrophages and CD8+ T-cells are given however, this can be insufficient to make conclusion about its role in immune system regulation.
7. Since NOX4 is NADPH oxidase, why a correlation between its expression and ROS and AOS was not studied?
8. English language grammar and style should be checked; for example, on line 31 “effective biomarkers”, lines 302, 309 (immune checkpoint), etc.
Checking and Editing are required
Author Response
Point 1: Title is suitable for review article, but not for research article. It is recommended to change it so to reflect the main results obtained in this study.
Response 1: The title of the article has been revised as “Identification of NOX4 as a new biomarker in hepatocellular carcinoma and its effect on sorafenib therapy”
Point 2: In the Abstract, the authors stated “Effectively biomarkers were necessary for HCC patient prognosis prediction and therapeutic target identification”. This sentence is confusing because it is necessary to distinguish the terms “prognosis” and “prediction”. Also, this difference should be discussed in the Introduction. Use the following papers: doi: 10.1016/j.ejca.2008.03.006 and doi: 10.1080/14737159.2021.1987217.
Response 2: “Effectively biomarkers were necessary for HCC patient prognosis prediction and therapeutic target identification”has been changed to “Novel prognostic biomarkers and therapeutic targets for HCC therapy are urgently needed to improve the survival of hepatocellular carcinoma (HCC) patients “in the abstract. The difference between“prognosis” and “prediction” has be discussed in the introduction section.
Point 3: The Introduction section is very poor, there is no discussion on roles NOX4 as NADPH oxidase and its implication in cancer. So, it is recommended to shortly discuss this using the following papers: doi: 10.3389/fcell.2022.884412; doi: 10.3389/fcell.2022.884412, etc.
Response 3: The role of NOX4 in different tumors has been discussed in the introduction section.
Point 4: Results section: in the second subsection the authors described the identification of hub genes, however, before this, Fig. 2 depicts relationships between clinicopathological characteristics and the hub genes – the logical presentation is disturbed.
Response 4: First, we used the GEO and Xena databases to identify 608 DEGs between HCC and normal tissues. Afterward we performed LASSO modelling on the TCGA expression matrix and selected eight genes (CDC20, NCAPG, CCNB1, MAD2L1, AURKA, TOP2A, NOX4, and CCNB2) for riskscore calculation. Then we used univariate logistic regression analysis to find the clinicopathological characteristics that may affect the prognosis of HCC patients, and combined them with the existing riskscore to construct a nomogram as shown in Figure 2.
Point 5: Fig. 2C – is there any correlation between DEGs and clinicopathological characteristics? Fig. 2E – are there survival rate for HCC patients or DEGs or hub genes?
Response 5: These clinicopathological characteristics were found by univariate logistic regression analysis, so there is no correlation between DEGs and clinicopathological characteristics. Figure 2E showed the survival analysis of 8 hub genes in HCC based on the TCGA data.
Point 6: Also, it is not clear, why on lines 306-317, the authors discussed the immune system and immune checkpoints – which of identified genes do regulate them? Also, on lines 470-472, results on the association of NOX4 expression and levels of macrophages and CD8+ T-cells are given however, this can be insufficient to make conclusion about its role in immune system regulation.
Response 6: The results revealed that NOX4 expression is positively associated with the immuno-suppressive molecule SIGLEC7 and negatively correlated with NK cells (Figure 3A, 3B). We further analyzed the TCGA-LIHC dataset and found that in the statistically significant NOX4 high expression group, the proportion of macro-phages is raised while the proportion of CD8+T cells is increased in NOX4 low expres-sion group (Figure 8E). Additionally, we examined the relationship between NOX4 expression and the effec-tiveness of immunotherapy in HCC using the Easlier package, and discovered that NOX4 low expression group had a greater percentage of immunotherapy response. (Figure 8F). All of the above results revealed immunotherapeutic significance of NOX4 in HCC. We will also test this in more cell and animal studies
Point 7: Since NOX4 is NADPH oxidase, why a correlation between its expression and ROS and AOS was not studied?
Response 7: With the ROS assay kit we found that overexpression of NOX4 was associated with higher ROS production and the results were shown in Figure 5I.
Point 8: English language grammar and style should be checked; for example, on line 31 “effective biomarkers”, lines 302, 309 (immune checkpoint), etc.
Response 8: English language grammar and style have been carefully reviewed and changes have been made accordingly in the new manuscript
Reviewer 3 Report
This is a single center retrospective study evaluating the impact of NOX4 expression in patients with hepatocellular carcinoma.I have several comments.
1. There are results written in Introduction section, this should be moved to results section.
2. In page 15, last line, "Figure 6D" should be "Figure 7D".
3. Why did the authors focused on NOX 4 in this manuscript? How did the authors define NOX4 high and low?
Author Response
Point 1: There are results written in Introduction section, this should be moved to results section.
Response 1: The introdection section has been changed as requested.
Point 2: In page 15, last line, "Figure 6D" should be "Figure 7D".
Response 2: Figure 6D has been changed to Figure7D in the new manuscript.
Point 3: Why did the authors focused on NOX4 in this manuscript? How did the authors define NOX4 high and low?
Response 3 : We first constructed an 8-gene prognostic model of HCC by means of bioinformation analysis. Then The mRNA levels of these genes were validated by qRT-PCR in 20 paired fresh HCC samples. We found that NOX4 was much more highly expressed in the HCC tissues than in the paracancerous tissues, while the expression of other genes differed inconsistently in cancer and the paracancerous tissues of different patients, as shown in Figure 3C. So we focused on NOX4 for further research.
We also performed immunohistochemical (IHC) staining to detect NOX4 protein expression on 150 paraffin-embedded HCC tissues from our hospital.The expression of NOX4 was assessed semi-quantitatively according to the intensity of staining (score 0, no staining; score 1, weak staining; score 2, moderate staining; score 3, strong staining) and the percentage of positive tumor cells (score 0, none; score 1, 1%-29%; score 2, 30%-69%; score 3, >70%). Multiplying the staining intensity score by the positive tumor cell score generated the overall immunohistochemistry score. The final score ranged from 0 to 9 and was interpreted as follows: negative (0), weak (1-3), moderate (4-6), and strong (>6). NOX4 expression was classified as high (grade 4-9) and low (grade 0-3).
Round 2
Reviewer 2 Report
Authors of the manuscript have provided the revised version. However, some concerns remain because some confusing statements are retained. So, the authors are recommended to make corrections and additional and more careful revisions.
1. In the Abstract, the revised statement sounds “Novel prognostic biomarkers and therapeutic targets for HCC therapy” – therapeutic … for HCC therapy is confusing.
2. Introduction section, a background for this study remains poorly provided. In the Introduction, proper discussion on NOX4 and its role as a target in cancer therapy REMAINS NOT PROVIDED. The recommended paper was ignored: doi: 10.3389/fcell.2022.884412 doi: 10.3389/fcell.2022.884412.
3. In the Introduction, the statement “identifying more reliable biomarkers to predict HCC prognosis” – to predict … prognosis is also confusing. Some biomarkers can be used aa both prognostic and predictive. However, the term “to predict” should be used in the context of treatment response in HCC patients, while the term “prognosis” should be used in the context of survival prediction independently of treatment.
4. In the Introduction, it is stated that “Reactive oxygen species are linked to the development of cancer and are produced by the NADPH oxidase (NOX) family [6]” – this should be corrected because there are many sources of ROS, among which NADPH oxidases are one of the major sources (not a single). Also, since NOX4 is proposed as a biomarker for treatment assessment, discuss the roles of predictive biomarkers using the paper: doi: 10.1080/14737159.2021.1987217.
5. Fig. 2E – on line 289, it is stated that “Patients with low risk had a better chance of surviving than those with high risk … (Figure 2E)”, however in the figure legend, it is stated “Survival analysis of 8 hub genes” – this is confusing, especially if take into account that TCGA resource enables survival analysis for patients that carry a definite upregulated/downregulated or mutated gene in a definite cancer type. How can the authors obtain a survival curve “FOR 8 GENES”? What is “high/low risk group” – with SIMULTANEOUS UPREGULATION/DOWNREGULATION OF 8 GENES? How it can be?
No comments
Author Response
Point 1: In the Abstract, the revised statement sounds “Novel prognostic biomarkers and therapeutic targets for HCC therapy” – therapeutic … for HCC therapy is confusing.
Response 1: In the abstract, this paragraph has been changed to “To improve the survival of patients with hepatocellular carcinoma (HCC), new biomarkers and therapeutic targets are urgently needed.”
Point 2: Introduction section, a background for this study remains poorly provided. In the Introduction, proper discussion on NOX4 and its role as a target in cancer therapy REMAINS NOT PROVIDED. The recommended paper was ignored: doi: 10.3389/fcell.2022.884412 doi: 10.3389/fcell.2022.884412.
Response 2: Proper discussion on NOX4 and its role as a target in cancer therapy has been added in the introduction section as suggested.
Point 3: In the Introduction, the statement “identifying more reliable biomarkers to predict HCC prognosis” – to predict … prognosis is also confusing. Some biomarkers can be used aa both prognostic and predictive. However, the term “to predict” should be used in the context of treatment response in HCC patients, while the term “prognosis” should be used in the context of survival prediction independently of treatment.
Response 3: In the Introduction, this pagragraph has been changed as “An 8-gene risk model was developed and its association with survival in HCC patients was evaluated.”
Point 4: In the Introduction, it is stated that “Reactive oxygen species are linked to the development of cancer and are produced by the NADPH oxidase (NOX) family [6]” – this should be corrected because there are many sources of ROS, among which NADPH oxidases are one of the major sources (not a single). Also, since NOX4 is proposed as a biomarker for treatment assessment, discuss the roles of predictive biomarkers using the paper: doi: 10.1080/14737159.2021.1987217.
Response 4: “Reactive oxygen species are linked to the development of cancer and are produced by the NADPH oxidase (NOX) family” has been corrected as “The membrane bound NADPH oxidases of the NOX family are a major source of ROS in cancer.” Besides, the roles of predictive biomarkers in HCC also discussed in the introduction section as requested.
Point 5: Fig. 2E – on line 289, it is stated that “Patients with low risk had a better chance of surviving than those with high risk … (Figure 2E)”, however in the figure legend, it is stated “Survival analysis of 8 hub genes” – this is confusing, especially if take into account that TCGA resource enables survival analysis for patients that carry a definite upregulated/downregulated or mutated gene in a definite cancer type. How can the authors obtain a survival curve “FOR 8 GENES”? What is “high/low risk group” – with SIMULTANEOUS UPREGULATION/DOWNREGULATION OF 8 GENES? How it can be?
Response 5: The selected genes were scored using the formular risk factor ∑ =∗ = riskScore Coef Expri i. The cut-off number was chosen using the rocr program, the expression of each gene was divided into high risk group and low risk group according to the selected threshold, so the patients were split into two groups based on their risk scores: high and low.
Reviewer 3 Report
The authors have revised the manuscript appropriately.
Author Response
Thank you for your review, we have revised the manuscript as required.